# Roles of Fascin in Dendritic Cells

**DOI:** 10.3390/cancers15143691

**Published:** 2023-07-20

**Authors:** Hao-Jie Wang, Ya-Ping Jiang, Jun-Ying Zhang, Xiao-Qi Tang, Jian-Shu Lou, Xin-Yun Huang

**Affiliations:** 1School of Pharmacy, Hangzhou Normal University, Hangzhou 311121, China; 2Key Laboratory of Elemene Class Anti-Cancer Chinese Medicines, Engineering Laboratory of Development and Application of Traditional Chinese Medicines, Collaborative Innovation Center of Traditional Chinese Medicines of Zhejiang Province, Hangzhou Normal University, Hangzhou 311121, China; 3Department of Physiology and Biophysics, Weill Cornell Medical College of Cornell University, New York, NY 10065, USA

**Keywords:** fascin, dendritic cells, tumor microenvironment, anti-tumor immune responses

## Abstract

**Simple Summary:**

Fascin is an actin-bundling protein that is highly expressed in mature dendritic cells and intratumoral dendritic cells. Dendritic cells are professional antigen-presenting cells of the immune system. Fascin is responsible for regulating dendritic cell migrations and tumor metastasis. The mechanism by which fascin regulates the functions of dendritic cells is currently being explored. In this review, we summarize fascin’s involvement in the physiological processes of normal dendritic cells including dendritic cell maturation, migration, and antigen presentation. In cancer patients, there is a subset of dendritic cells with high levels of fascin; activation of this subset of dendritic cells has been shown to enhance antitumor immune response. Fascin inhibitors increase intratumoral dendritic cell accumulation and activation, and cooperate with immune checkpoint inhibitors.

**Abstract:**

Dendritic cells (DCs) are professional antigen-presenting cells that play a crucial role in activating naive T cells through presenting antigen information, thereby influencing immunity and anti-cancer responses. Fascin, a 55-kDa actin-bundling protein, is highly expressed in mature DCs and serves as a marker protein for their identification. However, the precise role of fascin in intratumoral DCs remains poorly understood. In this review, we aim to summarize the role of fascin in both normal and intratumoral DCs. In normal DCs, fascin promotes immune effects through facilitating DC maturation and migration. Through targeting intratumoral DCs, fascin inhibitors enhance anti-tumor immune activity. These roles of fascin in different DC populations offer valuable insights for future research in immunotherapy and strategies aimed at improving cancer treatments.

## 1. Introduction

Dendritic cells (DCs) are the most efficient antigen-presenting cells of the immune system and play a crucial role in inducing primary immune responses and activating naive T cells [1,2,3,4]. DCs arise from hematopoietic progenitors and undergo myeloid and lymphoid differentiation pathways [5]. These cells exist in two states, immature and mature, with distinct functions [6]. Immature DCs are responsible for internalizing and processing antigens in non-lymphoid tissues, while under antigen stimulation, DCs mature and migrate to lymphoid tissues, thereby activating naive T cells and exerting immune effects [6,7].

Fascin possesses a unique structure comprising four beta-trefoil domains [8,9,10,11,12]. Fascin crosslinks F-actin filaments into parallel bundles and reorganizes the actin cytoskeleton, leading to the regulation of cell adhesion, migration, and cellular interactions [13,14,15]. Actin cytoskeleton remodeling provides the driving force for various cellular processes including cell migration and invasion [16,17]. Fascin bundles F-actin filaments through its two main actin-binding domains [18]. Fascin has been shown to participate in the formation of various membrane protrusions including filopodia, invadopodia, and veil-like protrusions [19]. Fascin depletion decreases the size and number of filopodia and invadopodia as well as affects actin dynamics [20].

Three fascin genes have been identified: fascin-1, fascin-2, and fascin-3. Fascin-1 is expressed in various types of cells, such as vascular endothelial cells, glial cells, and DCs, while fascin-2 and fascin-3 are predominantly found in retinal photoreceptor cells and the testis, respectively [21]. Fascin-1 (referred to as fascin here) is discussed in this review. Fascin is highly expressed in mature DCs but not detectable in immature DCs, macrophages, neutrophils, T cells, and B cells [22], making it a reliable marker for mature DCs. Additionally, fascin is overexpressed in various metastatic tumors and aberrantly expressed in different cancer types, where it is considered a marker and therapeutic target for cancer metastasis [23,24,25]. Fascin expression has been associated with tumor cell migration, invasion, metastasis, and adaptive immune response [24,25,26].

DCs are present both in normal tissues and the tumor microenvironment. When DCs migrate from peripheral tissues to the tumor microenvironment, they could uptake tumor antigens and undergo maturation, accompanied by increased fascin expression. Studies have shown increased fascin expression in mature DCs [6]. The upregulation of fascin in mature DCs enhances their migration [6,27,28,29]. DCs are highly motile cells, and fascin deficiency impairs their motility. Fascin plays a critical role in enhancing DCs’ motility through actin cytoskeleton reorganization (see below for details) [30,31,32,33]. Moreover, reduced fascin expression hampers cytoskeleton rearrangement in DCs/Langerhans cells, resulting in decreased T- cell activation [34]. Immature DCs lacking fascin exhibit high efficiency in antigen uptake but poor T cell activation [29].

Importantly, while high fascin expression in normal tissue DCs promotes immune responses, its overexpression in intratumoral DCs is associated with tumor development [35,36]. Immune cell profiling and proteomic analyses have demonstrated that inhibiting fascin expression in intratumoral DCs enhances antigen uptake and increases immune cell presence, suggesting that suppressing fascin activity contributes to tumor inhibition within the tumor microenvironment [37]. In addition, fascin inhibitors can impede intratumoral DC migration, leading to increased DC accumulation within the tumor microenvironment and promoting intratumoral DC activation [37]. There is a cell-intrinsic antagonism between DC migration and antigen uptake [36]. Within the immune process, DCs in the tumor microenvironment play a crucial role in fostering T cell immunity and immunotherapy responses, making them potential anti-tumor targets [38]. Fascin inhibitors can promote T cell activation through activated DCs, thereby achieving anti-tumor effects [37].

This review focuses on elucidating the role of fascin in normal and intratumoral DCs from an immune-related perspective, providing an understanding of how fascin regulates the functions of DCs.

## 2. Roles of Fascin in Normal Dendritic Cell Physiology

### 2.1. Fascin in DC Maturation

During the development of immature DCs into mature DCs, cytoskeletal changes take place [38]. Fascin is not expressed or is expressed at low levels in immature DCs but is highly expressed in mature DCs. Fascin is specifically induced and expressed during DC maturation, contributing to the dynamic assembly of veil-like membrane protrusions, the disassembly of podosomes, migration to lymph nodes, and the assembly of the immunological synapse [22].

Fascin promotes the transformation of actin filaments into actin bundles, leading to the formation of membranous protrusions on the surface of mature DCs, facilitating their circulation [22]. In vitro studies have shown that DCs lacking fascin genes exhibit a low occurrence of membrane-protrusive activities and impaired chemotaxis toward CCL19, a chemokine involved in mature DC lymphocyte recirculation [28]. Knockout of the fascin-1 gene in mice results in thinner, more spread DCs with fewer and smaller dorsal folds [22]. Re-expressing fascin through transfection can reverse the effects of fascin deletion [22]. Furthermore, inhibition of fascin expression using antisense constructs delays the morphologic maturation of DCs [38]. Thus, fascin contributes greatly to dendrite formation and the morphological maturation of DCs.

Furthermore, the fascin promoter exhibits robust activity in mature DCs and is transcriptionally targeted to mature DCs. The activity of the fascin promoter increases significantly during DC maturation. The regulatory sequence of the fascin promoter is located at the 5’-flanking promoter region, which contains a putative GC box, a composite cAMP responsive element/AP-1 binding site, and a TATA box [4]. High-level expression of fascin is induced during the maturation of DCs from CD14^+^ blood precursors in culture. Inhibition of fascin expression hampers the morphological maturation of DCs [27]. Fascin and other markers of mature DCs increase during the maturation of DCs. However, in mature DCs, knockdown of fascin has no impact on other markers of mature DCs, e.g., CD86, CD11c, and MHC II (major histocompatibility complex class II) [22].

### 2.2. Fascin in DC Migration

Fascin plays a crucial role in the maturation of DCs and promotes their migration. The maturation of DCs is characterized by the assembly of numerous membranous protrusions and the disassembly of podosomes [28]. Fascin is closely associated with the decomposition of podosomes, which are actin-rich membrane protrusions that act as cell matrix adhesion structures and hinder DC migration [28,39,40]. High levels of fascin weaken the ability of DCs to form podosomes [28]. Mature DCs lacking fascin exhibit low membrane activity and do not decompose podosomes [28]. However, after transfection with GFP-fascin, most DCs express fascin, decompose podosomes, and show increased migration to lymph nodes compared to fascin-knockout DCs [28]. When mature DCs lack fascin proteins, podosomes do not decompose [22]. DCs transiently lose podosomes shortly after activation, then recover podosomes within hours. Later, they permanently lose podosomes after activation, concomitant with the generation of characteristic veil-like membrane protrusions [28]. While the first and transient loss of podosomes is controlled via pathways involving prostaglandin E2, RhoA, rho-kinase, and ADAM17, fascin has been shown to increase the second and cause permanent loss of podosomes in mature DCs [39,41,42]. Hence, during DC maturation, the expression levels of fascin increase, and podosomes in these mature DCs undergo several rounds of growth and contraction before finally decomposing, which enhances their mobility [33]. The migration of mature DCs to lymph nodes facilitates subsequent antigen presentation processes.

### 2.3. Fascin in Antigen Presentation

Antigen presentation refers to the process through which antigen-presenting cells take up antigens, process them, and present immune peptides on their surface, which are then recognized by immunocompetent cells [43]. The reorganization of the actin cytoskeleton in DCs promotes antigen uptake, processing, presentation, and the activation of resting T cells [44]. How fascin is directly involved in antigen uptake and presentation is not clear. Immature DCs, with no or low levels of fascin proteins, have high antigen uptake activity. Mature DCs have high levels of fascin proteins and increased migration to the lymph nodes. It is not clear whether these mature DCs maintain high levels of fascin proteins after they reach the lymph nodes. Fascin has been shown to participate in the interactions between DCs and T cells. Regulatory T (Treg) cells suppress the function of DCs via direct physical contact using DC’s own fascin-dependent cytoskeleton, leading to weaker interactions between DCs and conventional T cells [45]. When fascin expression levels in DCs were reduced using siRNA, Treg cells could no longer bind DCs [45]. On the other hand, inhibiting fascin expression using antisense oligonucleotides reduced the heterologous stimulating activity of cultured bone-marrow-derived DCs. This decrease in fascin expression weakens the activation ability of DCs towards T cells [29] (Figure 1). Furthermore, CD40-CD40L signaling upregulates fascin expression in DCs during the activation of T cells, promoting continuous contact between DCs and T cells [46]. Additionally, ectopic expression of fascin can restore the contact between DCs and T cells and complete the antigen presentation process even in the absence of CD40-CD40L binding in DCs [46].

## 3. Roles of Fascin in Intratumoral DCs

### 3.1. High-Fascin Intratumoral DCs

Recent advancements in single-cell RNA sequencing (scRNA-seq) studies have significantly contributed to our understanding of the tumor microenvironment in cancer patients. Specifically, these studies have shed light on the upregulation of fascin expression in intratumoral DCs [47,48,49,50]. In a comprehensive scRNA-seq analysis of non-small-cell lung cancer (NSCLC) patients, DCs isolated from tumor tissues exhibited higher levels of fascin compared to matched non-tumor lung tissues. This distinct population of high-fascin expressed intratumoral DCs was termed “mregDCs” (mature DCs enriched in immunoregulatory molecules) [47]. Additionally, this subset of DCs showed elevated expression of CCR7, CCL19, LAMP3, and CCL22 [47]. Importantly, the activation of these specific mregDCs was found to enhance the response to anti-PD-1 antibody treatment, highlighting their potential role in immunotherapy [47]. It is worth noting that both cDC1 and cDC2 have the capacity to differentiate into mregDCs [47].

Furthermore, a comparative scRNA-seq study across colorectal, lung, ovarian, and breast cancers, including tumor and matched normal tissues, revealed higher expression of fascin in intratumoral DCs compared to other tumor-infiltrating immune cells, endothelial cells, and fibroblasts [48]. The intratumoral DCs were further categorized into five distinct phenotypes based on their transcriptomes: cDC1, cDC2, migratory cDCs, plasmacytoid DCs, and Langerhans-like DCs [48]. Among these subsets, migratory cDCs exhibited elevated expression of fascin, along with CCR7, CCL17, CCL19, and CCL22 [48].

Moreover, a scRNA-seq study focusing on hepatocellular carcinoma identified three enriched DC subsets: cDC1, cDC2, and LAMP3^+^ DCs [49]. The LAMP3^+^ DCs displayed high expression of LAMP3, fascin, CD80, CD83, CCR7, and CCL19 [49]. Importantly, their activation was found to enhance T cell-mediated cancer immunotherapy [49].

In another scRNA-seq investigation involving NSCLC tumor tissue samples, four distinct subsets of DCs were identified: cDC1, cDC2, pDC, and “activated” DCs [50]. The “activated” DCs were characterized by elevated expression levels of fascin, LAMP3, CCL19, CCR7, and CCL22 [50]. Notably, fascin’s increased expression defined this subset of “activated” DCs in both human and mouse lung tumor tissues [50].

Considering the consistent upregulation of fascin, CCR7, CCL19, CCL22, and LAMP3 among the different DC subsets identified by various research groups (mregDCs, migratory DCs, LAMP3^+^ DCs, and “activated” DCs), it is plausible that these subsets may represent the same or overlapping populations of DCs. Furthermore, the activation of these DC subsets has been shown to enhance the anti-PD-1 antibody antitumor immune response, suggesting their potential as targets for immunotherapeutic strategies [47].

### 3.2. Fascin Inhibition Increases Intratumoral DC Accumulation and Activation

Intratumoral DCs play a crucial role in obtaining tumor antigens, migrating to lymph nodes, and activating T cell responses [51,52]. The processes of antigen uptake and migration in DCs are antagonistic [36]. Fascin promotes DC migration by increasing podosome disassembly and reducing adhesion force [28]. Thus, fascin inhibitors decrease the podosome decomposition and the migration ability of DCs to leave tumor tissues, consequently leading to increased numbers of intratumoral DCs [22,37]. This was confirmed by the recent study that inhibition of fascin with a fascin inhibitor NP-G2-044 blocked the migration of intratumoral DCs [37], and fascin-deficient DCs showed low mobility compared with normal DCs [28]. Furthermore, the combination of NP-G2-044 and immune checkpoint inhibitor anti-PD-1 antibody increased the expression of CD40, CD80, and CD86, which are the markers of activated DCs [37]. The stimulatory molecule CD40 promotes T cell activation, regulates CD8^+^ T cell expansion, and enhances DC survival rate [53]. In addition, activation of DCs leads to the activation of CD8^+^ killer T cells in the tumor tissue, ultimately enhancing the anti-tumor immune response [37]. Activated intratumoral DCs upregulate T cell activation to initiate adaptive anti-tumor immunity [54,55]. Thus, fascin inhibitors lead to the accumulation of intratumoral DCs.

### 3.3. Fascin Inhibitors Cooperate with Immune Checkpoint Inhibitors to Enhance Anti-Tumor Immune Response

The immune checkpoint molecule PD-1 is widely expressed in intratumoral T cells [53]. Interaction between PD-1 and its ligand PD-L1 reduces the proliferation and effectiveness of CD8^+^ T cells [56]. The mechanisms through which fascin inhibition increases the anti-cancer immune cell response include the accumulation of DCs within the tumor tissues and the increased antigen uptake. First, in mouse studies, the number of intratumoral DCs is increased because of the reduced mobility of intratumoral DCs after inhibiting fascin [37]. Second, in both cell-based and mouse studies, fascin inhibitor treatment increased antigen uptakes by DCs [37]. Proteomic mass spectrometry analysis of proteins within the tumor microenvironment revealed that treatments with a fascin inhibitor and an anti-PD-1 antibody, compared with anti-PD-1 treatment alone, increased the levels of proteins involved in vesicle-mediated transport and clathrin-mediated endocytosis [37].

Furthermore, activated DCs produce interleukin 12 (IL-12), a pro-inflammatory cytokine produced by intratumoral DCs, which enhances the anti-tumor immune response [57]. Intratumoral DCs sense interferon γ (IFN-γ) released from neighboring T cells, leading to IL-12 production, which stimulates anti-tumor T cell immunity [58,59]. Prolonged activation of DCs results in lower IFN-γ secretion from T cells, and deficiencies in IL-12 affect T cell activation [59]. This leads to the induction of a T cell depletion program and the formation of an immunosuppressive microenvironment [60,61]. However, inhibition of fascin, in combination with anti-PD-1 antibodies, can relieve this inhibition, reactivate DCs, and initiate immune responses that inhibit tumor aggressiveness [37]. The contributions of IL-12 and IFN-γ to the anti-tumor effect of fascin inhibition and anti-PD-1 antibodies were verified using neutralizing anti-IL-12 and IFN-γ monoclonal antibodies [37]. Neutralization of IL-12 and/or IFN-γ eliminated the increased overall survival effect observed with the combination of the fascin inhibitor and anti-PD-1 antibody [37]. Hence, targeting high-fascin intratumoral DCs is crucial to enhance the anti-tumor immune response of anti-PD-1 antibodies (Figure 2).

## 4. Conclusions

Fascin plays a crucial role in the maturation, migration, antigen uptake, and antigen presentation of DCs. Mature DCs with highly expressed fascin migrate to lymph nodes via the decomposition of podosomes, the decrease in adhesion, and the increase in membrane protrusion dynamics. Intratumoral DCs with a high expression of fascin are not conducive to enhancing anti-tumor immune response. The number of intratumoral DCs is increased through reducing the mobility of intratumoral DCs after inhibiting fascin, and activated DCs are increased when used with anti-PD-1 antibody, further activating T cells. The detailed molecular mechanisms through which fascin inhibition enhances the immune response require further investigation.

## Figures and Tables

**Figure 1 cancers-15-03691-f001:**
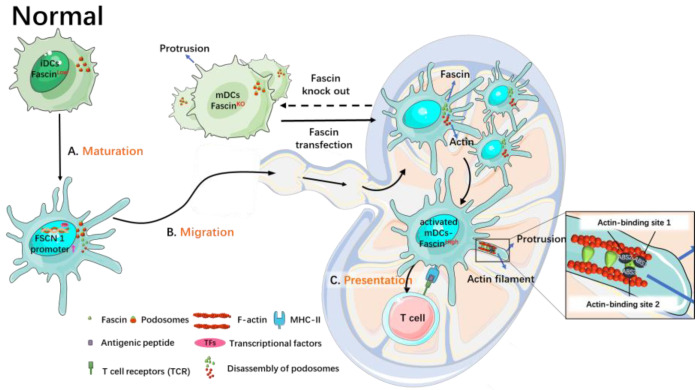
Roles of fascin in the maturation, migration, and antigen presentation of normal DCs. Fascin is barely expressed, if at all, in immature DCs and is induced during their maturation, leading to dynamic assembly of veil-like membrane protrusions, breakdown of podosomes, and migration to lymph nodes. This process results in an increased number of DCs migrating to lymph nodes. Mature DCs lacking fascin exhibit lower membrane activity, undecomposed podosomes, and a thinner and more widely distributed morphology. Transfection of fascin protein restores fascin expression, decomposes podosomes, and enhances DC migration.

**Figure 2 cancers-15-03691-f002:**
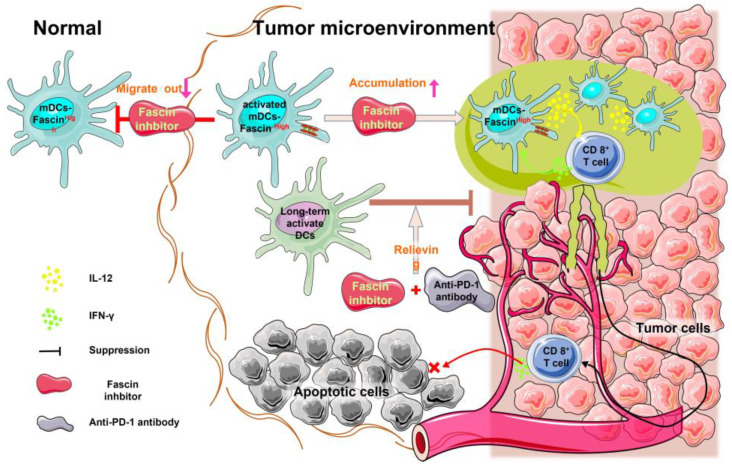
Inhibition of fascin decreases intratumoral DC migration and promotes DC accumulation within the tumor microenvironment. Migration and antigen uptake in DCs are antagonistic processes. High fascin expression in intratumoral DCs promotes their migration. Conversely, fascin inhibitors promote antigen uptake by these DCs, inhibit migration, and increase accumulation in the tumor microenvironment. Activated DCs secrete IL-12, which activates CD8^+^ T cells to secrete IFN-γ and exert an anti-tumor effect.

## Data Availability

Not applicable.

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
