# Peer review of "Roles of Fascin in Dendritic Cells"

_cancers, 2023, doi:10.3390/cancers15143691_

Round 1
Reviewer 1 Report
Wang et al., reviewed the role of Fascin in dendritic cells (DCs) with focus on the impact of DCs on tumor cell defense. They summarize that Fascin-1 is overexpressed in many types of cancers, and explain the role of Fascin in a sup-population of DCs. In these DCs a high Fascin level facilitates disassembly of podosomes and assembly of cellular protrusions. Also, the authors state that Fascin promotes DC migration by the providing adhesion force through podosomes and increases MHC-II expression. In the last chapters, the authors focus on the role of Fascin-1 expression on intratumoral DCs, and discuss the apparently contractionary finding that Fascin-1 inhibition increases the T-cell response (in particular in combination with PD-1 inhibitors) to malignant tumor cells.
This topic is very relevant, and the manuscript is well written. However, there are some issues that have to be addressed:
The role of F-actin in actin dynamics is not introduced. It should be briefly mentioned that remodeling of the actin cytoskeleton is essential for cellular migration, invasion etc. and that Fascin bundles F-actin because it has two actin binding domains. Furthermore, it should be clarified that this actin bundling activity is essential for the formation of different kind of protrusions; such as invadopodia, filopodia etc. Here, a cartoon is very helpful.
In addition, there are a lot of wrong citations (see below).
Detailed issues:
Page 1 .line 40: Please delete “protein” ; Fascin, a 55-kDa, and ………use “an” unique
Line 44: “Fascin-1 is expressed in various types of cells”, please mention the cell-types or at least the tissue. Here, citation [13] does not fit. The authors could cite this paper: https://www.ncbi.nlm.nih.gov/pmc/articles/PMC2324166/. Also expression of Fascin-2 and -3 has to be specified.
Page 2, line 57-59: “enhancing DC’s motility “. Since the mechanism is explained later, add (for details, see below).
Page 2, line 92: “Knockout of Fascin”; the authors should specify, knockout in mice?
Page 3, line 124: “increased expression of MHCII”. Do the authors mean gene expression or cell surface presentation? Again the citations [32, 38] do not fit.
Page 5, line 188: “a fascin inhibitor”; please mention the inhibitor, there are several Fascin inhibitors.
Line 199: “cause the accumulation of” . Please explain the mechanism.
The authors do not precisely explain the mechanism by which Fascin inhibition increases the anti-cancer immune cell response. The authors even write that Fascin inhibition “relieve this (T-cell) inhibition” (page 5 ,line 2013). It is counterintuitive that Fascin is essential for dendritic cell function but its inhibition increases the immune cell response. Wang et al., 2021 assume that the Fascin expressing DCs chronically stimulate T-cells resulting in their exhaustion. For the reader it will be hard to understand why inhibition of a protein essential for DC function increases the anti-tumor response of immune cells. Thus, this has to explained in detail.
Also, the mechanism of how Fascin promotes migration is not clearly explained; on the one hand the authors state that Fascin increases podomsome disassembly (page 3, line 105). On the other hand they state that "Fascin promotes DC migration by providing adhesion force through podosomes “. This has to be clarified.
Lastly, there is a nearly complete copy and paste section on page 4 line 155 from of the Discussion part of Wang et al. 2021 “ In recent years…….”
Reviewer 2 Report
The review summarizes the role of the protein fascin in dendritic cells (DCs) and in cancer.
The authors particularly focus on its dual main functions: on one hand its expression promotes DCs maturation and expression of proteins and receptors involved in T cell activation and antigen presentation (MHC molecules, CD86 and others); on the other hand, its expression promotes also DC migration to lymph nodes thanks to the induced decomposition of podosomes.
The authors present and review available data and studies related to these two functions and their involvement in cancer. However, the information, although rather encompassing, is presented mostly as a list of facts, disjointed and not critically analyzed. The authors report that inhibition of fascin expression facilitates DC permanence at cancer site leading to improved antigen presentation and therapeutic outcomes. However, the fact that this same inhibition should impact negatively also on the expression of DC maturation markers is not even mentioned or critically analyzed. The authors mention instead that inhibition of fascin expression in intratumoral DCs increase the expression of activation markers (line 192). This observation is in contrast with what reported in section 2.1 and should be thus critically discussed.
Data presented from different studies should be more critically discussed and put into perspective, similar to what is done in section 3.1 (lines 178-182) but lacking or should be improved in the rest.
The authors propose also that fascin inhibition can constitute a valid therapeutic option against cancer. How would this therapy be administered? How to ensure selectivity towards intratumoral DCs (intratumoral injection ?) versus systemic effects? How was fascin inhibition achieved in the studies presented and how the authors suggest this could be translated in the clinic?
We thus suggest further review of this manuscript in a more critical manner, prior to publication.
Line 113-114: the authors mention two distinct decomposition processes. Can the authors give some further insights on the mechanisms and differences between them.
Line 127: what do the authors mean by “unidirectional” ?
Figure 2: what the authors refer to as antigenic peptides I think should be more correctly called TCR receptor; peptides are presented loaded on MHC molecules at DC surface and interact and are recognized by T cell surface receptors.
Ref 30 reports two distinct studies, needs revision.
Round 2
Reviewer 1 Report
The manuscript has been clearly improved, I have no further comments
Reviewer 2 Report
The authors have addressed all raised points. We reccomend threfore publication of this manuscript in its current form.